# Distinct Optical and Structural (Nanoyarn and Nanomat-like Structure) Characteristics of Zinc Oxide Nanofilm Derived by Using *Salvia officinalis* Leaves Extract Made without and with PEO Polymer

**DOI:** 10.3390/ma16134510

**Published:** 2023-06-21

**Authors:** Adnan H. Alrajhi, Naser M. Ahmed, Mohd Mahadi Halim, Abeer S. Altowyan, Mohamad Nurul Azmi, Munirah A. Almessiere

**Affiliations:** 1School of Physics, Universiti Sains Malaysia, Penang 11800, Malaysia; adnan.alrajhi@yahoo.com (A.H.A.); nas_tiji@yahoo.com (N.M.A.); mmhalim@usm.my (M.M.H.); 2Department of Physics, College of Science, Princess Nourah Bint Abdulrahman University, P.O. Box 84428, Riyadh 11671, Saudi Arabia; 3School of Chemical Sciences, Universiti Sains Malaysia, Penang 11800, Malaysia; mnazmi@usm.my; 4Department of Biophysics, Institute for Research & Medical Consultations (IRMC), Imam Abdulrahman Bin Faisal University, P.O. Box 1982, Dammam 31441, Saudi Arabia; mmussar@gmail.com; 5Department of Physics, College of Science, Imam Abdulrahman Bin Faisal University, P.O. Box 1982, Dammam 31441, Saudi Arabia

**Keywords:** ZnO NPs, ZnO NRs, *Salvia officinalis*, PEO, structures, optical properties

## Abstract

This paper reports the optical properties of zinc oxide nanofilm fabricated by using organic natural products from *Salvia officinalis* leaves (SOL) extract and discusses the effect of the nanocrystal (NC) structure (nanoyarn and nanomat-like structure) on nanofilm optical properties. The surface-active layer of the nanofilm of ZnO nanoparticles (ZnO NPs) was passivated with natural organic SOL leaves hydrothermally, then accumulated on zinc oxide nanorods (ZnO NRs). The nanofilms were fabricated (with and without PEO) on glass substrate (at 85 °C for 16 h) via chemical solution deposition (CSD). The samples were characterized by UV-vis, PL, FESEM, XRD, and TEM measurements. TEM micrographs confirmed the nucleation of ZnO NPs around 4 nm and the size distribution at 1.2 nm of ZnO QDs as an influence of the quantum confinement effect (QCE). The nanofilms fabricated with SOL surfactant (which works as a capping agent for ZnO NPs) represent distinct optoelectronic properties when compared to bulk ZnO. FESEM images of the nanofilms revealed nanoyarn and nanomat-like structures resembling morphologies. The XRD patterns of the samples exhibited the existence of ZnO nanocrystallites (ZnO NCs) with (100), (002), and (101) growth planes. The nanofilms fabricated represented a distinct optical property through absorption and broad emission, as the optical energy band gap reduced as the nanofilms annealed (at 120 ℃). Based on the obtained results, it was established that phytochemicals extracted from organic natural SOL leaves have a distinct influence on zoic oxide nanofilm fabrication, which may be useful for visible light spectrum trapping. The nanofilms can be used in photovoltaic solar cell applications.

## 1. Introduction

An inevitable future vision of clean energy is meeting the rising energy demand for long-term usage, wherein an environmentally friendly material with a green preparation method is needed [1]. Because of the difficulties of conventional approaches in obtaining a new eco-friendly material source for the demands of renewable energy generation, the use of organic natural products is gaining much attention [2]. The impressive green synthesis of ZnO NPs nanofilms could be used in optical and electrical applications. These applications include photovoltaics, optoelectronics, solar cells, etc. [3,4,5]. Bulk ZnO suffers from a high exciton (electron-hole pair) recombination rate and has a wide band gap of about 3.37 eV [6]. The properties of ZnO NPs serve as intermediaries between atomic molecules and bulk components, making them useful for widespread applications [7]. In addition, ZnO NPs have garnered much interest because of their wide energy band gap (3.37 eV) and high electronic mobility (115–155 cm^2^ V^−1^ s^−1^). These characteristics are greatly suitable for electronic transport and reduce the loss of photo-generated carrier recombination [8]. In addition, the size dependence of ZnO NPs increases the electron trapping rate as cluster size increases [9]. The ultra-small ZnO quantum dots (ZnO QDs) show promising optoelectronic properties compared to their bulk counterparts [10,11]. At this dimension, the quantum confinement of ZnO NPs takes place, leading to carrier confinements [12]. As nanoparticles reach the size of quantum dots, optical properties are influenced, and they have a larger surface-to-volume ratio with a high extension coefficient [13]. The carriers (valence and conduction bands) lead to localized states [12]. In addition, charge carrier (electrons and holes) motion is confined in three dimensions, which is optically active in the nanometric-size space region [12]. Whereas optical absorption enhances the free carrier generation and separates it spatially, thus increasing nanofilm conductivity [14]. 

Repeated studies have shown that surface-passivated ZnO NPs work as efficient semiconductor nanocrystals (SNCs), providing emergent properties that may be beneficial for the growth of distinct active layer performance optoelectronic nanofilms and photovoltaic devices [15]. The green synthesis of organic products from plant extracts that passivate the nanoparticle surfaces can form crystalline nanoparticles (NCs) and thus act as an active layer in the nanofilm, giving the active layer the opportunity to trap light and potentially transfer the photocurrent [16]. Essentially, the nanofilm surface containing zinc oxide nanocrystals (ZnO NCs) may ameliorate electron transit and extraction, ensuring efficient electron transmission. ZnO NCs may be useful in a broad array of uses, such as efficient light-emitting devices, solar cells, and so forth [1,2,9,17]. The remarkable formation of trapped states at the interface of metal oxide semiconductors and inhibition of the carrier recombination due to surface passivation give ZnO NCs potential for diverse uses in optoelectronics [18,19,20,21]. Previously, most of the preparation techniques for ZnO NCs have involved some protecting agents to improve the nanofilm stability and optoelectronic properties [19,20,21]. ZnO NCs with distinct scales and geometries are predicted to play a vital role in light spectrum absorption in the blue-green wavelength range. The obtained ZnO NCs with antireflection and light absorption traits allow the tailoring of various device performances in photovoltaic applications [16]. This fulfills the main priorities, such as enhanced light trapping with increased light absorption, reduced incident of light reflection, and modified optical response of the nanofilms desired for various applications [16,22,23]. In the past, different natural products derived from the leaves of Aloe barbadensis Miller, Moringa oleifera, Calotropis gigantea, Neem, and Justicia procumbense have been used to make ZnO NPs [23,24,25,26,27].

Various green synthesis routes have recently been developed to produce different semiconductor nanocrystallites (SNCs) without harsh conditions or expensive or harmful chemicals. The selection of proper capping ligands plays a significant role in determining the various physical and chemical properties of SNCs [7,11,28]. Furthermore, plant-mediated synthesis has the added advantage of producing ZnO NCs that can be coated with organic phytochemicals [28,29]. In addition, it is possible to modify organic surfactants by keeping the proper dispersion of ZnO NPs together with polymers that efficiently hinder the agglomeration of NPs [30]. The addition of polymers into the mixture can maintain the ZnO NPs stability in the polymer matrix, wherein the polymers cluster together and act as stabilizing agents for the ZnO NPs, thus preventing self-aggregation when preparing the ZnO NPs [18,31]. In brief, the surfactant (SOL) considerably improves the optical properties of ZnO NPs by preventing their agglomeration [9,31,32]. Several factors, such as surfactant content, pH, and temperature, can be tuned to optimize the growth evolution and dispersion of the colloidal ZnO NPs. By connecting the phytochemical component (as a capping agent) to the surface of the ZnO, we can appreciably reduce the size of the ZnO QDs [18]. It has been established that the use of organic natural products enclosing phytochemicals as surfactant capping agents, such as *Salvia officinalis* leaves (SOL) (without and with polymers such as polyethylene oxides (PEO)), can provide a successful linkage between the technology and eco-friendly resources. Furthermore, this may provide some impetus for the easy production of stable and pure (impurities-free) environmentally friendly commercial products without the need for much energy, complex synthesis protocols, elevated pressure, or noxious chemicals [10,20,21]. The originality of this study lies in fabricating ZnO NP nanofilms by hydrothermal chemical solution deposition (CSD) using organic natural products (SOL leaves) through an eco-friendly method using DW. The literature represents the usage of different metals and fabrication layers [33,34], with a new approach to ZnO NPs (nanoyarn and nanomat-like structures). 

In this study, after much research effort, efficient nanofilms containing ZnO NPs as an active layer with surface passivation were fabricated using a green synthesis approach (with or without PEO). Considering the basic and applied importance of surface-passivated ZnO NPs, this study adopted an easy and efficient green synthesis strategy to fabricate nanofilms from tiny ZnO NPs passivated with organic natural SOL leaf extract [31]. The fabricated nanofilms were characterized using different analytical tools to determine their structure, morphologies, and optical characteristics, and they promised to have a distinct optical characterization. Fabricated nanofilm may be used for photovoltaic applications, such as solar cells.

## 2. Experimental Procedures

### 2.1. Samples Preparation

In this study, ZnO NPs were green synthesis passivated with organic natural products extracted from *Salvia officinalis* (SOL) leaves (purchased from the local market of the Sultanate of Oman), with and without polyethylene oxides (PEO, R & M Chemicals with Batch Number PBME290119) (purity of 99%), mixed with zinc acetate [$C_{4}H_{6}O_{4}\mathrm{Zn},2H_{2}O$] (Merck KGaA 64271, Darmstadt, Germany) (purity of 99%), and hexamethylenetetramine [$C_{6}H_{12}N_{4}$] Scharlab, Spain (CAS: [100−97-0]) (purity of 99%). 100 nm ZnO nanofilm was fabricated on a glass substrate (seed layer) using the Auto HHV500 Sputter Coater or Radio Frequency (RF) and Direct Current (DC) Sputtering Magnetron Sputtering. Meanwhile, the ZnO NRs were grown on the seed layer by mixing 500 mM zinc acetate dehydrate and 500 mM hexamethylenetetramine in a final volume of 100 mL of distilled water (DW), with constant stirring for an hour. The same glass substrate was then immersed in the solution containing ZnO seeds at an angle of 45° for 5 h at 85 °C (Figure 1). 

The ZnO NRs were used to resolve the issue of the low optoelectronic activation of ZnO NPs in nanofilms. The ZnO NRs extended from the ZnO seeds (interface) up into the active layer of the ZnO NPs. This ensured the separation of the excitons in both the lower and upper portions of the active layer, where the excitons can easily move to the adjacent ZnO NRs, resulting in photon path assistance [34,35]. Thus, the one-dimensional (1D) nanorods (ZnO NRs) facilitated the transport of charge carriers with their remarkably reduced recombination [3]. Additionally, the presence of ZnO NRs prevented charge recombination and made both diffusion length and electron lifetime longer, thereby enhancing electron transport [36,37]. Figure 2 shows a FESEM image of the corresponding ZnO seed layer and ZnO NRs. The accumulation of ZnO NPs on the ZnO NRs influenced the optical properties (Figure 3).

The ZnO NPs acted as an active layer on the top of the nanofilm, the active layer trapped the light ranges (UV-visible) with the appropriate ZnO NPs optical band gap energy and reduced the light-scattering effect, while the ZnO NRs enhanced the light-harvesting properties [3]. The absorbance spectrum of the ZnO NRs was in the UV-vis region, the optical band gap energy ($E_{g}$) of ZnO NRs was (3.51 eV), Figure 4, which was just near the bulk ZnO band gap of (3.37 eV) [6], and the photoluminescence PL emission spectrum of ZnO NRs showed low intensity, as noticed in Figure 5. 

### 2.2. Active Layer Synthesis

The SOL extracts were prepared using the Soxhlet apparatus, which involved approximately 12 g of pulverized SOL leaves in filter paper (Whatman filter paper purchased from Sigma Aldrich) and 200 mL of distilled water (DW) put in a 250 mL flask, followed by boiling for 4 h to a translucent appearance. The obtained solution was cooled down to ambient temperature and stored at 10 °C (Figure 6). ZnO NPs were green synthesized at room temperature by mixing equal amounts of zinc acetate dehydrate and hexamethylenetetramine (500 mM) in DW under continuous stirring for 1 h to get a final volume of 200 mL. The mixture was then divided evenly (each of 100 mL) between two beakers. Subsequently, 3 mL of SOL and 0.5 g of PEO were added to the first beaker to make a composition of ZnO NPs with SOL phytochemical extracts and PEO (ZnO NPs/SOL-PEO). In the second beaker, 3 mL of SOL (without PEO) was taken to make a composition of ZnO NPs with SOL (ZnO NPs/SOL). Both mixtures were thoroughly stirred at room temperature by a magnetic stirrer. ZnO NRs (prepared on the glass substrate) were immersed at an angle in the individual beakers that contained ZnO NPs/SOL-PEO and ZnO NPs/SOL (Figure 6). The beakers were kept in the chemical solution deposition (CSD) chamber at 85 °C for 16 h. The obtained nanofilms were cleaned with DW and dried using an air blower for further characterization.

## 3. Results and Discussion

### 3.1. Optical Absorbance and Morphology of Colloidal ZnO NPs 

Optical properties of the ZnO NPs were characterized by using UV-Vis-NIR absorption spectra, and photoluminescence emission was captured using the Agilent Carry 5000 absorption spectrophotometer. The PL spectrometer (Liconix 3205 N) coupled to an HPC-2 light source collimation system was utilized to record the PL spectra (325–1000 nm), respectively. As ZnO NPs have a wide band gap energy (3.37 eV) with UV-visible light absorption [8,31], organic capping agents were used from the extracted organic natural products (SOL); these include phytochemicals such as flavonoids and phenolic acids, including apigenin, ferulic acid, quercetin, and luteolin; and phenolic acids, including methyl rosmarinate, rosmarenic acid, cinnamic acid, caffeic acid, quinic acid, and chlorogenic acid [38,39,40,41,42,43]. Those organic phytochemicals can interact with the inorganic ZnO and cause changes in the optical properties of nanoparticles [44]. The polyphenols in *Salvia officinalis* combine with ZnO NPs as a result of nanoparticle capping, which improves optical properties. Optical band gap energy can be reduced through this interaction, allowing blue-visible light spectrum absorption [44,45,46,47]. As aforementioned, the choice of a proper surfactant could improve the optical band gap of ZnO NPs by preventing their agglomeration tendency [33]. With the addition of SOL surfactant to the mixture, the phytochemical components were tied to the Zn surface, forming ZnO NPs. As PEO was added to the mixture, interactions between ZnO and PEOs resulted in passivating ZnO NPs surfaces; the hydroxyl groups located on the ZnO surface interacted with the carboxyl groups in the polymer chain [48], enhancing the quantum confinement effect (QCE) of ZnO NPs [48]. In addition, the volume-to-size interface dominated the whole volume of the composite as the size of the ZnO NPs decreased [49,50,51,52].

In the region of 290–320 nm, the UV–Vis spectra of the ZnO NPs displayed a broad, noticeable absorption band. The literature absorption band of ZnO NPs at approximately 317 nm revealed the excitonic nature of the samples [31]. The inset depicts the Tauc plot, employed to determine the optical band gap energy ($E_{g}$) of the Zn NPs [48,50,53,54]; Figure 7a,b, (with/without PEO, respectively). Both samples showed absorbance falls as the wavelength was raised. The larger electronic density of states in the ZnO NPs was the likely reason for the higher transition probability, which in turn can be attributed to the higher absorption intensity of the ZnO NPs. 

The value of $E_{g}$ for ZnO NPs with PEO inclusion was about 3.31 eV, while the value of $E_{g}$ for ZnO NPs without PEO inclusion was approximately 3.16 eV. The observed decrease in the energy absorbance value for ZnO NPs as PEO was excluded can be attributed to the NP’s magnification to a larger size. The emission optical band gap energy ($E_{g}$) values of the ZnO NPs were calculated using the formula $E_{g}=\frac{hc}{\lambda }$, where *c* represents the speed of light and *h* is the Planck constant [7]. The ZnO NPs were excited by the 380 nm source, shown by the blue peak in Figure 8, as some UV-visible light from the source may have passed through the sample and was not all absorbed [10]. PL emission intensity peak of ZnO NPs colloidal about 490 nm and the corresponding optical band gap energy ($E_{g}$) is 2.53 eV (with/without PEO).

As depicted in Figure 8, the PL results indicate that SOL phytochemicals had a significant effect on the emission of suspended colloidal ZnO NPs in the blue-green region extending to the yellow-orange region [55]. The broadened emission spectrum may have been due to the various band gaps in the colloidal ZnO NPs and ZnO capping agents of different sizes that absorb various wavelengths and so emit in various wavelengths, which is supported by the TEM results (Figure 9b,e) [31]. Figure 8a,b shows two examples of aqueous solution syntheses (with and without PEO, respectively). The ZnO NP-prepared samples almost showed the same intensity peak (about 490 nm) due to particle size controlled by the reduction and capping of SOL phytochemicals (with/without PEO). In addition, the ZnO NPs were protected from aggregation after being treated with SOL extracts (with/without PEO) [31].

An energy-filtered transmission electron microscope (EFTEM, Libra 120, Zeiss GmbH, Oberkochen, Germany) was used to examine the morphologies of colloidal ZnO NPs. Figure 9a–c displays the TEM image of ZnO NPs/SOL-PEO, NP size distribution, and sizes of several ZnO NPs with SOL shells (with PEO). The morphology clearly verified the QCE of the ZnO NPs by SOL phytochemicals. The size distribution of the ZnO NPs was around 13.92 nm to 24.90 nm, and the histogram plot showed that the maximum ZnO NP diameter was around 3 nm, wherein tinier ZnO NPs were more abundant in the colloidal suspension. The agglomeration of ZnO NPs was prevented in both samples by capping the shell around the ZnO NPs core. Figure 9d–f shows the TEM image micrograph of ZnO NPs/SOL (without PEO); the sized particles ranged from 4.84 nm to 3.67 nm, and the histogram plot indicates a maximum diameter distribution of 1.2 nm, and the ZnO NP size distribution of several ZnO NPs with SOL capping shells (excluding PEO).

The QCE refers to the physical phenomenon of nanoparticles, in which a material’s electrical characteristics change as its size decreases to the nanoscale. When ZnO NPs are lowered to the nanoscale, their band gap increases considerably, resulting in unique optical and electrical characteristics [31]. Several phytochemicals derived from *Salvia officinalis*, including polyphenols and flavonoids, have been shown to enclose and stabilize ZnO NPs [31]. SOL leaves extracts are useful in regulating the size and morphology of ZnO NPs during synthesis, which is required to achieve the QCE. Furthermore, SOL extract is useful in reducing ZnO NP aggregation, as SOL phytochemical is assigned to the surface of ZnO NPs [44,56]. The phytochemical (phenolic and flavonoid), (Figure 10a,b), functional groups (chlorogenic acid, rosmarinic acid, ellagic acid, 7-glucoside, and luteolin) may be incorporated on the surface of ZnO NPs and are responsible for ZnO NP stabilization [46,57,58]. 

Those components are an example of many *Salvia officinalis* components that may bind to ZnO NPs, as well as having the ability to cap with ions and induce ZnO nucleation. Flavonoid components initiate the capping of ions, whereas phenolic compounds form multi-chelating bonds and stabilize the ZnO NPs after nucleation, resulting in the formation of various sized nanoparticles [7,44]. 

### 3.2. Structure, Morphology, and Optical Characteristics of Nanofilms

#### 3.2.1. FESEM and XRD Analysis

Using an energy-dispersive X-ray energy diffraction (EDX, FEI Nova SEM 450, FEI Company, Hillsboro, OR, USA) spectrometer, the elemental compositions of the samples were identified. A field emission scanning electron microscope (FESEM, FEI Nova SEM 450, FEI Company, Hillsboro, OR, USA) was used to examine the morphology of the samples. Figure 11a,b displays the FESEM images with the corresponding EDX spectra (Figure 11c,d) of ZnO NPs/SOL-PEO and ZnO NPs/SOL nanofilms (with/without polymer), respectively. The surface morphology of the ZnO NPs/SOL-PEO nanofilms showed the nucleation of ZnO NCs composed of a new nanostructure (nanoyarn and nanomat-like) makeup. The dimension of the nanoyarn structure was determined using the Image J software program (version 1.53 k), with a nanoyarn diameter of 14.2 μm and a width of 133 nm. When PEO was excluded, the nanofilm showed the nanoyarn-like structure again as an indicator that this structure belongs to the SOL leaf extract phytochemicals, the nanoyarn had a diameter of 25 μm wherein the flower-like structure was embedded in the nanoyarn-like morphology. The cross-section images of the ZnO NPs/SOL-PEO and ZnO NPs/SOL films (with/without PEO) showed an effective accumulation of the nanoyarn structure (Figure 12a,b). These results were attributed to the influence of SOL phytochemicals on the structural rearrangements. As revealed by the EDX spectral analysis (Figure 12c,d), the EDX showed the amount of Zn (Wt% 80.61) with PEO and (Wt%: 79.33) without PEO. In brief, the use of SOL surfactant could have preserved the Zn amount as its natural product capping agent (Table 1).

The non-destructive characterization technique of X-ray diffraction (XRD, Bruker D8 Advance, AXS GmbH, Karlsruhe, Germany) was used to identify information about the structure and crystalline phases of nanomaterials, such as the preferred crystallite size and orientation. Table 2 shows the positions of various significant XRD peaks with their corresponding lattice orientations for the ZnO NPs/SOL-PEO and ZnO NPs/SOL nanofilms (without and with annealing at 120 °C). Figure 13 depicts the XRD profiles of all the studied nanofilms. The XRD profiles displayed sharp crystalline peaks that corresponded to the JCPDS card number 36−1451 of the hexagonal wurtzite lattice structure of the ZnO crystal [10]. The sharpness of the XRD peaks with high intensity indicated a large crystallite size (strong crystallinity) [5]. The XRD patterns of the nanofilms further confirmed the presence of such lattices [31]. 

The ZnO NPs/SOL-PEO and ZnO NPs/SOL nanofilms showed intense diffraction peaks (Figure 13) at $31^{o}$, $34^{o},$ and $36^{o}$, corresponding to the lattice planer orientations of (100), (002), and (101), respectively. The most intense (002) peak indicated the preferred growth direction of the nanocrystallites; Figure 13a (with SOL and PEO) represents both capping agents (SOL and PEO) having strong crystallinity along the (002) orientation. When PEO was excluded from the mixture, the orientations (100) and (101) had strong crystallinity (Figure 13b). As annealing was applied (2h at 120 °C, increasing gradually), sharp XRD peaks were increased at the orientations (100) and (101) at both the ZnO NPs/SOL-PEO and ZnO NPs/SOL nanofilms; Figure 13c,d, respectively. Indicating an improvement in nanofilm crystallinity (sharp intensity), thermal annealing for a prolonged time allowed better crystallization. The sharper XRD peaks might have been due to the easy migration of Zn and O atoms into the lattice and their subsequent occupation at the vacant sites, allowing a significant reduction in the lattice due to annealing [4,5,60]. The Debye-Scherrer equation was used to calculate the particle size, wherein the most intense XRD peak at (002) was selected [20]: (1)$$D=\frac{K\lambda }{\beta Cos\theta }$$
where θ is Bragg’s angle, β is the full width at half maximum (FWHM) of the intense XRD peak, λ is the X-ray wavelength (0.15406 nm), and K = 0.89 is a constant. 

Table 3 represents the crystallite size of different plane orientations, with more crystallinity (sharp peaks) at the (100), (002), and (101) orientation planes of the nanofilms. Annealing modifies the crystallinity of ZnO nanofilm and has a significant effect on the size distribution of the ZnO nanoparticles, the crystal quality, and the presence of surface defects. The nanofilms showed a decrease in crystal size from 58.42 nm to 43.82 nm in the (100) orientation plane and 59.13 nm to 44.34 nm in the (101) orientation plane (with/without PEO, respectively). This decrease in crystal size may have been attributed to the absence of PEO chains, as the hydroxyl groups located on the ZnO surface interacted with the carboxyl groups in the polymer chain [46,47]. In addition, the polymer chain connected to the SOL phytochemicals, which may explain the enlargement in the ZnO NPs/SOL-PEO nanofilm and the reduction in crystal size as PEO was excluded. The (002) orientation had a crystal size of 58.42 nm and 58.83 nm (with/without PEO, respectively). The same (002) crystal orientation showed a slight increase in crystal size as annealing was applied. Where there was no remarkable change in crystal size at the (100) orientation plane before and after annealing for both nanofilms (with/without PEO). The (101) orientation plane showed a decrease in crystal size of ZnO NPs/SOL-PEO nanofilm (with PEO) from (59.13 nm to 44.34 nm); this was because annealing reduces but does not eliminate native defects, optimizes crystal structure, and reduces oxygen vacancy [61,62]. Whereas ZnO NPs/SOL nanofilm (without PEO) showed an increase at (101) crystal orientation from (44.34 nm to 59.12 nm), in addition to the effect of phytochemicals in ZnO NCs nucleation, the annealing indicated that the ZnO NCs transformed from their original crystal structure to a more robust crystal structure, with an increase in crystallinity and a decrease in the local atomic defects of oxygen vacancy, during the annealing technique. Overall, crystallinity changed in response to annealing, indicating that this process significantly affects nucleation and crystal growth [61]. The average crystal lattice for different orientation planes in the ZnO NPs/SOL-PEO nanofilm was 59.12 nm, while with PEO excluded, the size was 56.86 nm. 

The XRD data analysis showed sharp nanocrystallinity (good crystallinity) in the nanofilms. This in turn produced microvoids, O or Zn vacancies [60,62,63] making some significant structural adjustments that were responsible for an improvement in the optical absorption and emission properties. During the annealing process, the atoms received sufficient energy and moved towards relative equilibrium locations, thus decreasing the lattice strain, improving the nanocrystallinity, and reducing the microvoids in the nanofilm [9,64]. In addition, thermal annealing caused an increase in the crystallite size, density, and oxygen absorption of the nanofilm, thereby increasing the Zn atomic ratio. These structural alterations, caused by thermal annealing of the nanofilms, played a vital role in the improvement of the morphology and optical properties of the nanofilms [64,65]. 

#### 3.2.2. Optical Characteristics of Nanofilm

The optical absorbance of the fabricated nanofilm represented absorbance in the blue-green region, and the optical band gap energy ($E_{g}$), evaluated by Tauc’s plot using the spectral absorption data of the nanofilms, was $E_{g}$ = 2.65 eV, while $E_{g}$ = 2.67 eV when PEO was excluded from the mixture; Figure 14a,b, respectively [66]. The active layer structure of the ZnO NCs may have effectively affected light absorbance. ZnO NCs (nanoyarn structure) may provide a trapping surface to the spectral absorbance of the white light spectrum and the distinct optical properties of the nanofilms [16,22,66,67]. These features and techniques are important for light trapping. Furthermore, the presence of nanoyarn structures could enhance the antireflection of light, minimize optical losses, and increase light absorption.

The active layers of ZnO NPs/SOL-PEO nanofilm and ZnO NPs/SOL nanofilm showed distinct optical properties, wherein the former nanofilms showed broad emission and a peak at 596 nm and 606 nm (with/without PEO, respectively). The optical band gap energy ($E_{g}$), calculated using the formula $E_{g}=\frac{hc}{\lambda }$, was equal to $E_{g}$ = 2.08 eV and $E_{g}$ = 2.05 eV (with/without PEO), with a higher intensity as PEO was excluded (Figure 15a,b, respectively). Conversely, the formation of ZnO NCs enclosing nanoyarn and nanomat-like morphologies may have acted as light trapping, increasing the surface area, and enhancing the antireflection of light; thus, the large ZnO NCs in the nanofilm outermost layers may offer an advantage in UV-visible white light absorbance and PL emission to the yellow-red region, which is related to the performance of the ZnO NC crystal quality in the nanofilm active layer [56].

#### 3.2.3. Effects of Annealing on Optical Properties of Nanofilms

The effect of annealing on the optical properties of the ZnO nanofilms was studied. Annealing changes the crystal quality of ZnO nanofilm, which has a considerable impact on the size distribution of the ZnO nanoparticles, crystal quality, the occurrence of surface defects, and optical characteristics. Due to the increased surface area of the developed nanofilm dangling bonds, atoms, and local atomic defects are formed at the surface texture rather than the internal cores [68,69]. The particle size, which was altered as annealing was applied, demonstrated that annealing had a significant impact on nucleation, crystal growth, and tiny grain aggregation. The crystallinity was evidenced by the sharp intensity in the XRD findings, and UV-visible and broad photoluminescence bands. Furthermore, annealing reduces but does not completely eliminate native defects, optimizes crystal structure, and decreases oxygen vacancies [62,70]. In addition, the optoelectronic characteristic nanofilms are affected by their crystallinity; the optical band gap energy of ZnO NPs decreases with annealing due to changes in their crystal structure. The phonon mode in conjunction with annealing indicates that the ZnO NCs transformed from their original crystal structure to a more robust crystal structure during the annealing technique, with an improvement in crystallinity (Figure 13c,d), and a decrease in the local atomic defects of oxygen vacancy. The optical properties (UV-visible and photoluminescence) and nanofilm band gap of the nanofilms were amended [62,71]. It was asserted that the thin film containing ZnO NCs was greatly efficient for light collection and photoconversion [13], which was mainly due to the large surface area to volume ratio and the high extinction coefficient of the NCs [69,72]. 

The optical band gap energy ($E_{g}$) of the annealed nanofilms was calculated by Tauc’s plot using the spectral absorption data of the nanofilms, represented by ($E_{g}$ = 2.53 eV) for ZnO NPs/SOL-PEO nanofilm, and ($E_{g}$ = 1.89 eV) (without PEO) for ZnO NPs/SOL nanofilm; Figure 16a,b, respectively. The optical band gap energy recorded after the nanofilms annealed was due to the modification in the active layer of ZnO NC crystallinity, as pre-discussed, the impact of annealing on the improvement in crystallinity, and a decrease in the local atomic defects of oxygen vacancy. The crystallinity became more sharp (Figure 13d) (without PEO) as an indication of good crystal formation after annealing [65,73], with an optical band gap energy of ($E_{g}$ = 1.89 eV), thus tailoring the optical properties of the nanofilms beneficial for the photoanode in solar cell applications [16]. Table 4 represents the optical band gap energies of the nanofilms fabricated before and after annealing. As annealing was applied, the electron-phonon interaction was boosted, and the number of phonons increased. Further to the crystal structure disorder (vacancies, crystal disorder, etc.), these two factors influenced the formation of absorption tails. In addition, thermal disorder and the exciton interaction with crystal lattice influenced the optical properties of nanofilm [74,75,76]. Figure 16 represents the optical band gap energy ($E_{g}$) variant with (αhν)2 of ZnO NPs nanofilm.

The PL emission of both thin films (ZnO NPs/SOL-PEO and ZnO NPs/SOL) showed high-intensity emission, with broad emission in the green-red regions. The emission optical band gap of the annealed nanofilms was calculated using the formula $E_{g}=\frac{hc}{\lambda }$ and the emission peaks at 609 nm and 612 nm were ($E_{g}$ = 2.04 eV) and ($E_{g}$ = 2.03 eV) (with/without PEO respectively; Figure 17a,b, respectively). In addition, the ZnO NPs/SOL-PEO nanofilm intensity emission increased after annealing (Figure 17a) compared to before annealing (Figure 15a), as annealing enhanced the nanofilm crystallinity [75,76]. Conversely, the formation of ZnO NCs enclosing the nanoyarn and nanomat-like morphologies may have acted as light trapping, increasing the surface area, and enhancing the antireflection of light. 

Figure 18a represents the effect of SOL extract leaf (with/without PEO) absorption and the PL of all the parameters used in this study. The absorption intensity increased with broader absorbance at (ZnO+SOL) ZnO NPs/SOL nanofilm compared to the absorbance at (ZnO+SOL+PEO) ZnO NPs/SOL-PEO nanofilm. PEO caused the double influence of phytochemical adsorption on ZnO as a capping agent with phytochemicals from SOL leaf extracts [50]. The FESEM image of ZnO NCs of ZnO NPs/SOL nanofilm (Figure 11), representing a nanoflower structure with nanoyarn and nanomat-like structures, may have increased the surface area of the nanofilm, increasing the possibility of light trapping, compared to the nanofilm with PEO. As discussed in this study, annealing has an impact on the nanofilm crystallinity, defects, nanoparticle distribution, etc. Therefore, the absorption was broadened (with/without PEO), with preference to ZnO NPs/SOL nanofilm, as annealing increased the crystallinity sharpness (good crystallinity), especially at (100) and (101) orientations (Figure 13d), compared to with PEO (Figure 13c). All these features affected the nanofilm optical band gap energy ($E_{g}$), with preference to the nanofilm fabricated without PEO, which had ($E_{g}$ = 1.89 eV) of ZnO NPs/SOL nanofilm compared to an ($E_{g}$ = 2.53 eV) of ZnO NPs/SOL-PEO for that fabricated with PEO; Figure 18b. 

## 4. Conclusions 

The optical, structural, and morphological properties of ZnO NPs/SOL-PEO and ZnO NPs/SOL nanofilms, fabricated using organic natural products extracted from SOL leaves, were examined. The green synthesis route was followed to make an active surface layer of the nanofilms. It was established that the surface passivation of ZnO NPs was made hydrothermally and was successfully coated with SOL phytochemicals. TEM micrographs confirmed the nucleation of the ZnO NP cores, which were capped by phytochemical shells (with/without PEO). The nanoyarn and nanomat-like structures of ZnO NCs may act as UV-visible light traps. The fabricated nanofilms indicated distinct optical properties with optical band gap energy of ($E_{g}$) of 2.65 eV and 2.67 eV (with/without PEO, respectively). The annealed nanofilms (at 120 ℃ for 2 h) showed the crystallinity evidenced by the sharp intensity in the XRD findings in (100) and (101) planar orientations, and band gap energy of ($E_{g}$) (2.53 eV) and (1.89 eV) with high emission intensity photoluminescence (with/without PEO, respectively). It was demonstrated in the study that the distinct optical characterization of the fabricated nanofilms using organic natural products extracted from SOL leaves can be used in photovoltaic applications, such as solar cells.

## Figures and Tables

**Figure 1 materials-16-04510-f001:**
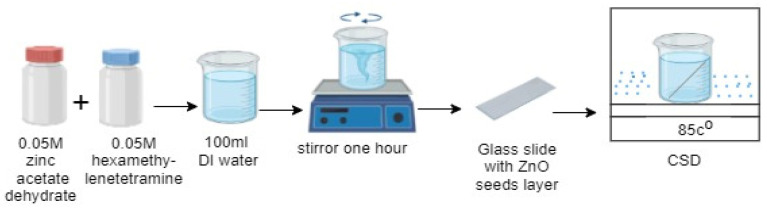
Steps of ZnO NRs preparation.

**Figure 2 materials-16-04510-f002:**
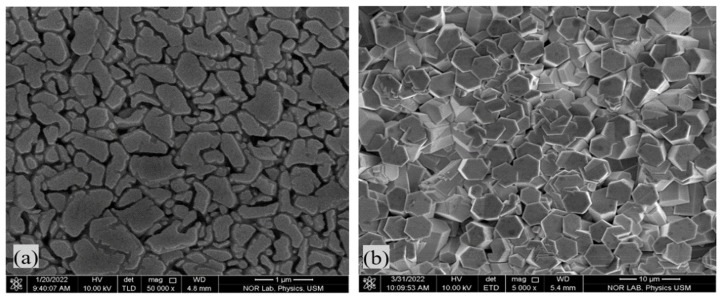
FESEM image micrographs of the (**a**) ZnO seeds layer, and (**b**) ZnO NRs.

**Figure 3 materials-16-04510-f003:**
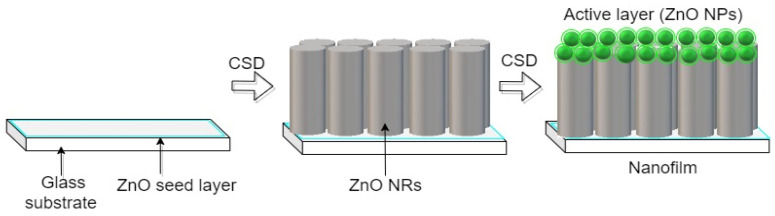
Mechanism of active layer formation, (ZnO NPs onto ZnO NRs stacking in the nanofilm).

**Figure 4 materials-16-04510-f004:**
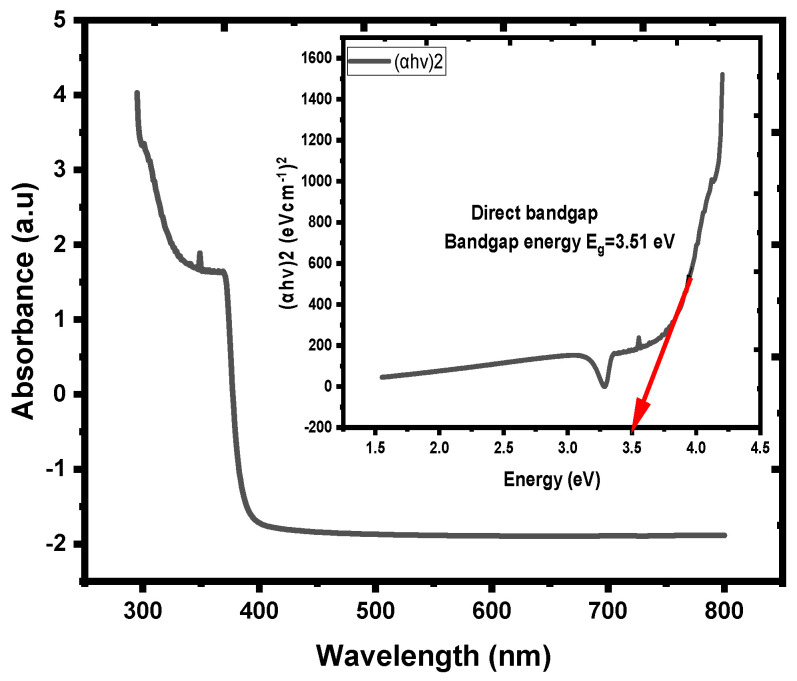
Absorption spectra of ZnO NRs. Insets contain the corresponding optical band gap energy ($E_{g}$).

**Figure 5 materials-16-04510-f005:**
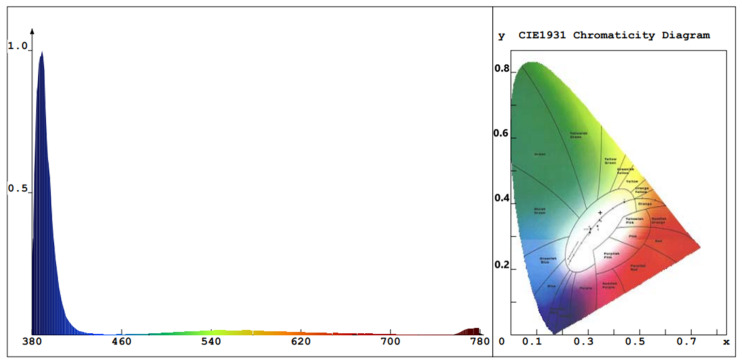
PL emission spectra of the proposed ZnO NRs. Insets contain the corresponding CIE diagram exhibiting color purity.

**Figure 6 materials-16-04510-f006:**
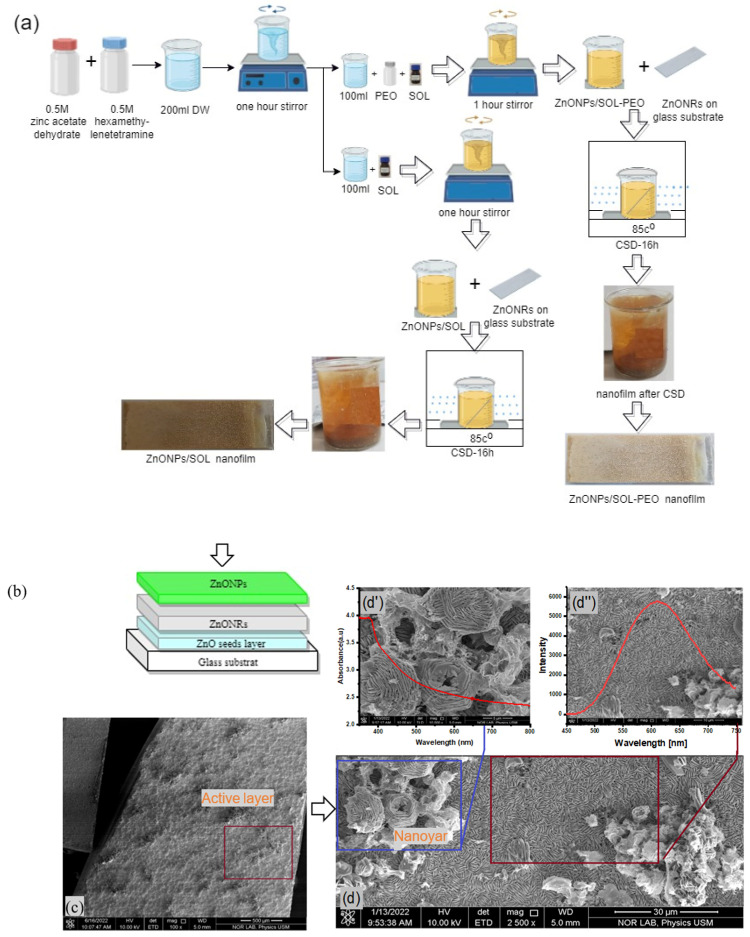
(**a**) Various stages of nanofilm fabrication, (**b**) illustration of ZnO NPs thin film; FESEM image micrograph of (**c**) ZnO NPs thin film (500 μm), (**d**) nanoyarn and nanomat-like structures (30 μm), (**d′**) optical absorbance of the film containing ZnO NPs (5 μm), and (**d″**) emission spectra of nanofilm with ZnO NPs (10 μm).

**Figure 7 materials-16-04510-f007:**
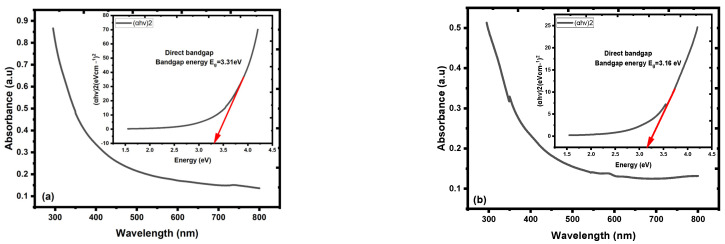
Absorption spectra of ZnO NPs coated with SOL capping agent: (**a**) with PEO, (**b**) without PEO. Insets contain the corresponding optical band gap energy ($E_{g}$).

**Figure 8 materials-16-04510-f008:**
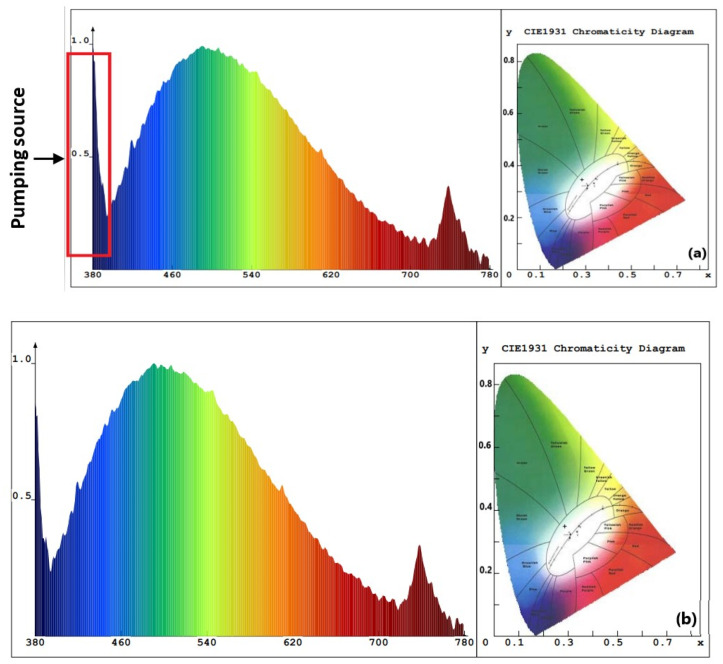
PL emission spectra of the proposed ZnO NPs synthesized with *Salvia officinalis* phytochemicals: (**a**) with PEO, (**b**) without PEO. Insets contain the circular CIE diagram exhibiting color purity.

**Figure 9 materials-16-04510-f009:**
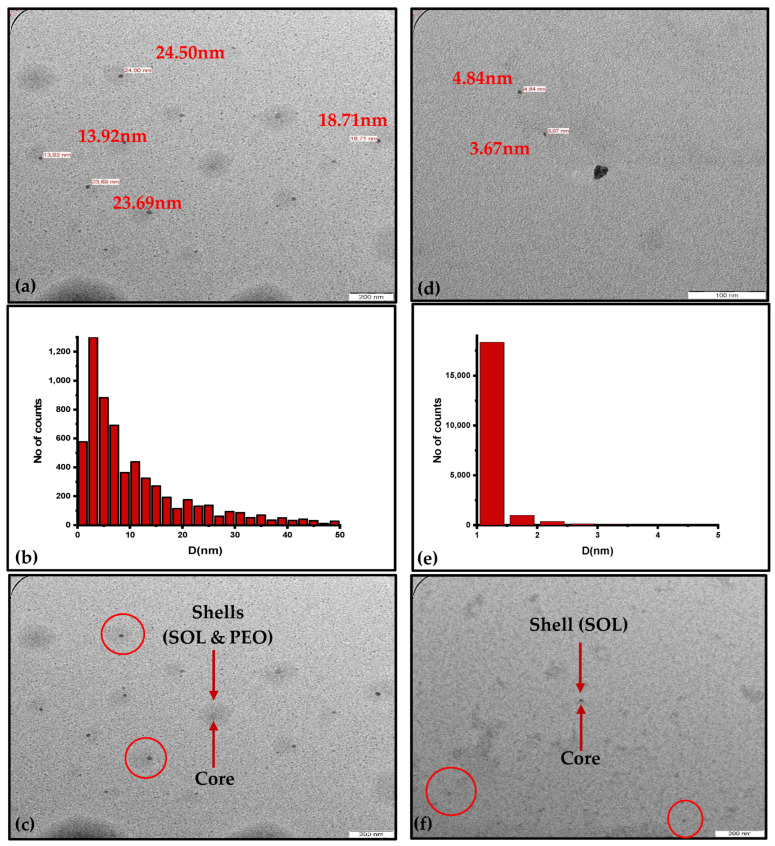
(**a**–**c**) TEM image of ZnO NPs/SOL-PEO: sizes of several particles, particle size distribution, and ZnO NPs capped with SOL + PEO shells, respectively. (**d**–**f**) TEM image of ZnO NPs/SOL: sizes of several particles, particle size distribution, and ZnO NPs capped with SOL shell, respectively.

**Figure 10 materials-16-04510-f010:**
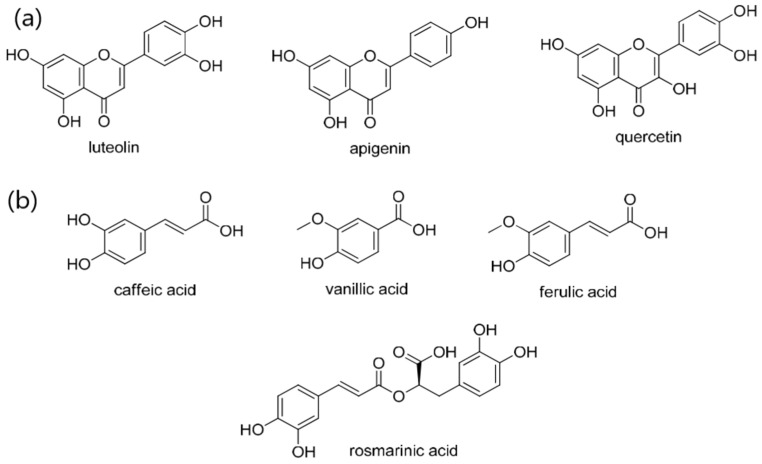
Phenolic components from SOL extract: (**a**) flavonoids [59], (**b**) phenolic acids [59].

**Figure 11 materials-16-04510-f011:**
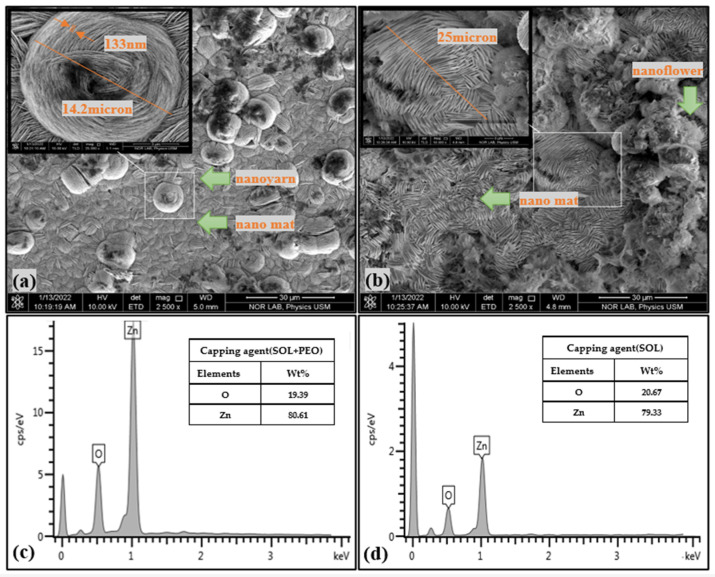
FESEM image of (**a**) ZnO NPs/SOL-PEO nanofilm, and (**b**) ZnO NPs/SOL nanofilm. EDX spectra of (**c**) ZnO NPs/SOL-PEO nanofilm, and (**d**) ZnO NPs/SOL nanofilm.

**Figure 12 materials-16-04510-f012:**
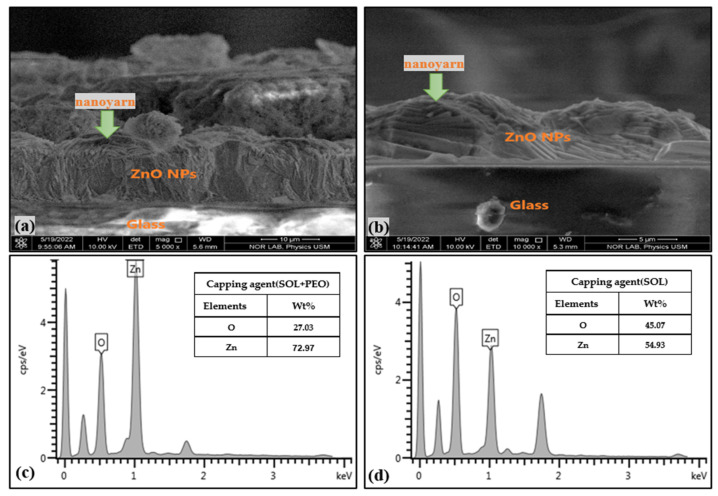
Cross-section view of (**a**) ZnO NPs/SOL-PEO nanofilm, and (**b**) ZnO NPs/SOL nanofilm. EDX spectra of (**c**) ZnO NPs/SOL-PEO nanofilm, and (**d**) ZnO NPs/SOL nanofilm.

**Figure 13 materials-16-04510-f013:**
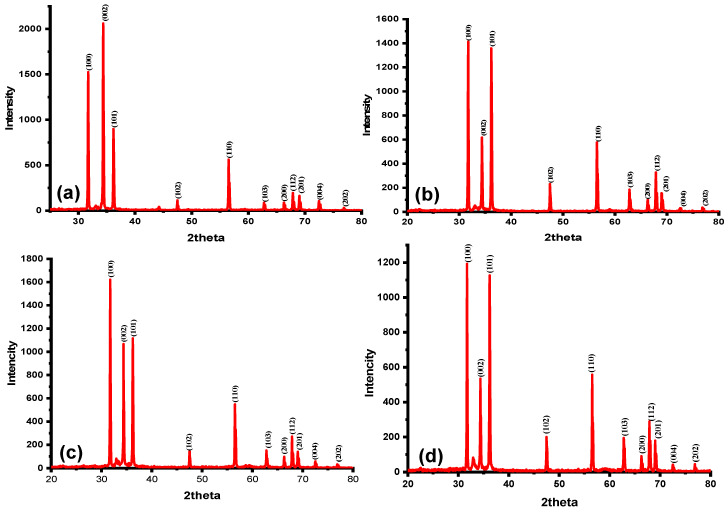
XRD profiles of nanofilms (**a**) ZnO NPs/SOL-PEO, (**b**) ZnO NPs/SOL, (**c**) ZnO NPs/SOL-PEO annealed at 120 ℃ for 2 h, and (**d**) ZnO NPs/SOL annealed at 120 ℃ for 2 h.

**Figure 14 materials-16-04510-f014:**
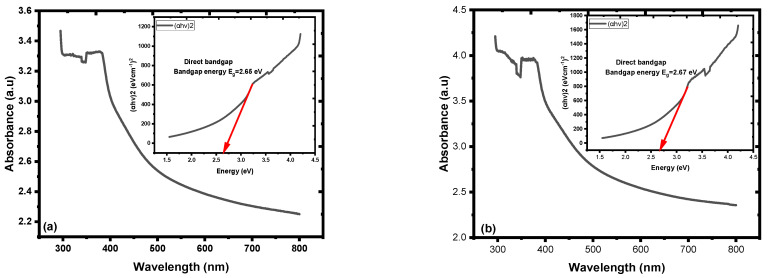
Absorption spectra of the nanofilms: (**a**) ZnO NPs/SOL-PEO nanofilm, and (**b**) ZnO NPs/SOL nanofilm. Insets contain the corresponding optical band gap energy ($E_{g}$).

**Figure 15 materials-16-04510-f015:**
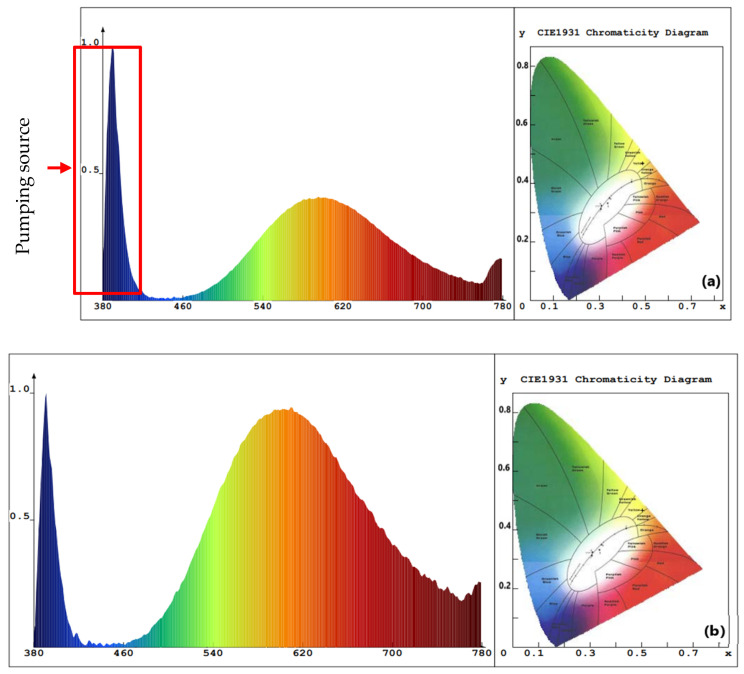
PL emission spectra of the nanofilms: (**a**) ZnO NPs/SOL-PEO nanofilm, and (**b**) ZnO NPs/SOL nanofilm. Insets contain the corresponding CIE diagram exhibiting color purity.

**Figure 16 materials-16-04510-f016:**
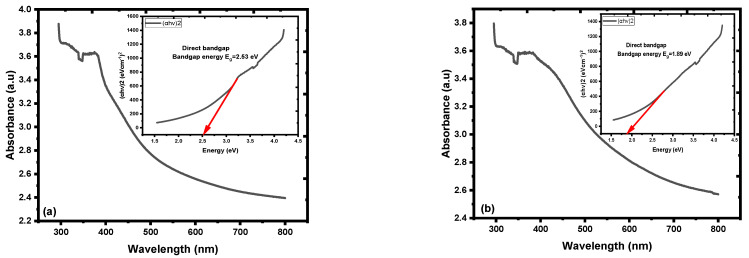
Absorption spectra of the annealed (at 120 ℃ for 2 h) nanofilms: (**a**) ZnO NPs/SOL-PEO nanofilm, and (**b**) ZnO NPs/SOL nanofilm. Insets contain the corresponding optical band gap energy ($E_{g}$).

**Figure 17 materials-16-04510-f017:**
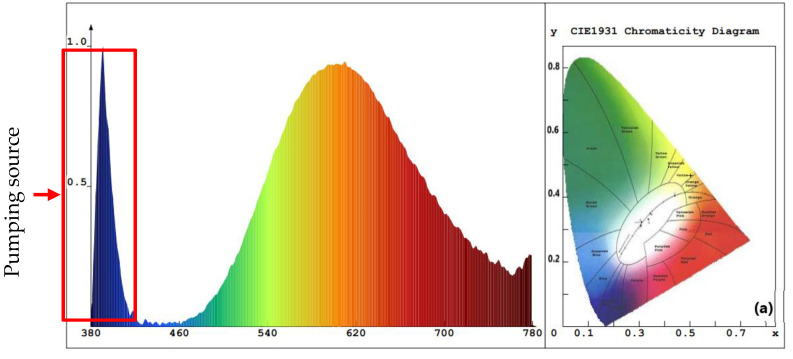
PL emission spectra of the annealed (at 120 ℃ for 2 h) nanofilms: (**a**) ZnO NPs/SOL-PEO nanofilm, and (**b**) ZnO NPs/SOL nanofilm. Insets contain the corresponding CIE diagram exhibiting color purity.

**Figure 18 materials-16-04510-f018:**
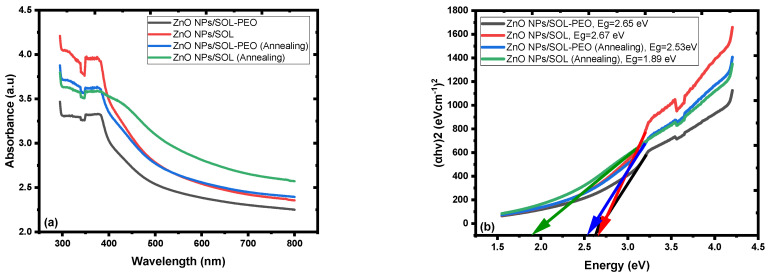
(**a**) Absorption spectra of various fabricated ZnO NP nanofilms. (**b**) Optical band gap energy ($E_{g}$) variant with (αhν)2 of ZnO NP nanofilms.

**Table 1 materials-16-04510-t001:** Zn and O elemental weight against capping agent obtained from EDX analysis.

Capping Agent	SOL + PEO	SOL
Element	Wt%	Wt%
Zinc	80.61	79.33
Oxygen	19.39	20.67

**Table 2 materials-16-04510-t002:** XRD peaks analysis of synthesized nanofilms.

Samples	XRD Peaks Position (in Deg.)	Corresponding Lattice Indices
ZnO NPs/SOL-PEO	31.7436, 34.3966, 36.2299, 47.5324, 56.5544, 62.8521, 66.3049, 67.838, 68.9822, 72.47	(100), (002), (101), (102), (110), (103), (200), (112), (201), (004), (202)
ZnO NPs/SOL	31.7373, 34.4055, 36.227, 47.5332, 56.5333, 62.7762, 66.2783, 67.8637, 68.9997, 72.5916	(100), (002), (101), (102), (110), (103), (200), (112), (201), (004), (202)
ZnO NPs/SOL-PEO annealed at 120 °C	31.7377, 34.4028, 36.2263, 47.5207, 56.5581, 62.7786, 66.2884, 67.8554, 68.9876, 72.581	(100), (002), (101), (102), (110), (103), (200), (112), (201), (004), (202)
ZnO NPs/SOL annealed at 120 °C	31.7352, 34.4009, 36.2248, 47.5274, 56.5288, 62.7752, 66.283, 67.8626, 68.989, 72.562	(100), (002), (101), (102), (110), (103), (200), (112), (201), (004), (202)

**Table 3 materials-16-04510-t003:** Crystallite size of nanofilms, calculated using the Debye–Scherrer equation.

	D (nm)			
hkl	ZnO NPs/SOL-PEO	ZnO NPs/SOL	ZnO NPs/SOL-PEO Annealed	ZnO NPs/SOL Annealed
1 0 0	58.42	43.82	58.42	43.82
0 0 2	58.42	58.83	58.83	58.83
1 0 1	59.13	44.34	44.34	59.12
1 0 2	46.05	46.05	61.4	46.05
1 1 0	47.86	38.28	38.29	47.85
1 0 3	49.4	49.38	65.84	49.38
2 0 0	50.35	67.12	67.12	67.31
1 1 2	67.73	67.74	67.73	40.64
2 0 1	68.19	68.2	68.19	68.19
0 0 4	85.71	34.87	34.86	34.78
Average (D)	59.12	56.86	59.98	51.59

**Table 4 materials-16-04510-t004:** Optical band gap energies of the nanofilms.

Samples	$E_{g}\;(\mathrm{eV})\;\mathrm{Room}\;\mathrm{Temperature}$	$E_{g}$ (eV) after Annealing at 120 °C
ZnONCs/SOL-PEO	2.65	2.53
ZnONPs/SOL	2.67	1.89

## Data Availability

Due to privacy and ethical restrictions, the data supporting the reported results of this study are unavailable for sharing. We acknowledge the importance of data sharing and encourage researchers to comply with data availability policies. However, in this particular case, we are unable to provide access to the data analyzed or generated during the study. We apologize for any inconvenience this may cause.

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
