# Peer review of "Distinct Optical and Structural (Nanoyarn and Nanomat-like Structure) Characteristics of Zinc Oxide Nanofilm Derived by Using Salvia officinalis Leaves Extract Made without and with PEO Polymer"

_materials, 2023, doi:10.3390/ma16134510_

Round 1

Reviewer 1 Report

This paper reports the structure and optical properties of salvia officinalis leaves (SOL) extract-derived zinc oxide (ZnO) nanofilms. Since it describes a green technology, it is worthy of publication. Nevertheless, the paper has a general conceptual flaw. The manufactured structure consists of a sputtered ZnO seed layer, a ZnO nanorods layer and a ZnO nanoparticle layer, the latter both deposited by chemical solution deposition (Figure 3). The optical properties of such complex systems are much better understood by investigation each of the layers separately and modeling the optical properties of the system as a multilayer system.

Other shortages are:

1) p. 2, row 53: the bandgap of bulk ZnO is missing.

2) p. 2, row 55: a critical size below which confinement effects occurs (to my knowledge about 20 nm) and the nature of quantum confinement (usually quantum confinement describes the spatial confinement of electron-hole pairs (excitons) in one or more dimensions by potential wells, an alternative origin is the spatial confinement of phonons in ZnO quantum dots) should be given.

3) The use of a Tauc plot for the evaluation of ZnO bandgap is common, but it is not reliable due to the presence of Urbach tails. Moreover, the authors suppose a direct allowed transition for the bandgap without providing any evidence.

3) The authors use adjectives enhanced (p.7, row 218, p.11 row 314, p.14 row 382) and improved (p. 1 row 21, p. 3 row 104, p. 7 row 233, p. 10, row 295&296, p.11 row 307, 308, 328, p. 14 row 382) without noting in comparison to what.

4) TEM images in Figure 7 are of poor quality.

5) The red shift of the band gap together with a broad emission peak does not prove the formation of excitons. Exitons induce narrow peaks in absorption and photoluminescence spectra. Such narrow peaks were not detected.

6) p. 8 row 243: Please give the version of the Image J program.

other shortcomings:

- change optoelectronic traits to optoelectronic properties

- change chemical bath deposition to the more common term chemical solution deposition

Author Response

Reply to reviewer 1

Reviewer 2 Report

The authors propose the green synthesis approach to obtain high quality ZnO NPs from the natural SOL extracts and subsequent fabrication of an active layer of the nanofilm promising for the photovoltaic devices making. The work fits well with the current trend towards new green technologies. The work is important as it enriches green synthesis methods for NPs making. ZnO NPs were characterized by different methods that helped to reveal the composition, structure and morphology of the ZnO NPs. Nevertheless, the manuscript has some drawbacks and requires major revision.

1. Characterization methods UV-vis, PL, FESEM, XRD and TEM should be described better (equipment models, manufactures etc.) as well as the magnetron sputtering process details.

2. Line 98: “influences” should be replaced by “influenced”

3. Graphical determination of the band gap energy Eg is quite questionable. For example, it is easy to verify that Fig. 13a’ and Fig. 13b’ show almost the same form of the absorption spectra. But the results of the Eg graphical determination differ significantly. Note, that there are no markable linear sections in Fig. 11 and Fig. 13 in order to make interpolation to zero absorption reliably. In any case, such interpolation will often be somewhat arbitrary and subjective. Furthermore, it is not clear why the authors chose (αhν)2 instead of (αhν)1/2 or any other electron transition model. Unfortunately, there is a common problem of the correct band gap determination. It would be better to exclude the Eg determination results from the revised manuscript. Surely, the publication will not lose its significance after such a change.

Author Response

reply to reviewer 2

Reviewer 3 Report

The following issues must be addressed:

1.       The authors must underline in the Introduction part what is new and innovative in this work compared with other papers (avoid the use of pronouns such as “we” “us” “I”);

2.       The chemicals purity must be provided;

3.       Lines 145-147 “Furthermore…” – there are no proves for your claims.

4.       Authors mention several times about “efficient light conversion” but there are no values about that?!

5.       EDX analysis – please provide the elemental mapping.

6.       The authors claim that “Zn content was improved due to thermal annealing” – this is unclear, please provide more details.

7.       Check ref 5;

Author Response

reply to reviewer 3

Reviewer 4 Report

Manuscript ID: materials-2360343

Title: Distinct optical characteristics of salvia officinalis leaves extract-derived zinc oxide nanofilm made without and with PEO polymer

The present manuscript details the structural and optical analysis of a zinc oxide nanofilm surface derived from a SOL extract. To produce the thin films, chemical bath deposition (CBD) was employed, and various characterization techniques were employed to assess the films' properties. The authors have demonstrated that the SOL surfactant-passivated films exhibit superior optoelectronic performance owing to a strong quantum confinement effect. The annealed films had a significantly reduced optical bandgap. Although this research study appears to have been executed effectively and has relevance in the field of optoelectronics, several issues must be addressed before the paper can be considered for publication.

1. The manuscript should be reviewed carefully for grammar and syntax errors, as there are several typing mistakes present. For example, in line 98, "are strongly influences" should be corrected to "are strongly influenced."

2. The authors should investigate the effect of sampling on the quantity of zinc oxide obtained from SOL extracts and report their findings.

3. Further explanation is needed to clarify why PEO leads to a decreased band gap. The authors could include supporting literature data to strengthen their argument.

4. To support their optical band gap derivation using Tauc's procedure, the authors should consider citing references such as Optical Materials 58 (2016), 51-60 and Applied Physics B 119 (2015), 273-279.

5. The authors must provide a detailed explanation of why the SOL is beneficial in obtaining stronger QCE in ZnO NPs.

6. In lines 284 and 285, the authors mention particle sizes of 0.7665 nm and 0.7664 nm. The authors should clarify how they determined the particle size with such precision and report the uncertainty of those values.

7. The effect of thermal treatment on the large decrease in the optical band gap (from 2.36 to 1.54 eV) should be explained and emphasized more in the abstract and conclusion sections.

Moderate editing of English language.

Author Response

reply to reviewer 4

Reviewer 5 Report

This paper describes a topic on the structures and optical properties of salvia officinalis leaves (SOL) extract derived zinc oxide (ZnO) nanofilm surface.
The active surface layer of the film containing ZnO nanoparticles (NPs) with nanorods (NRs) activation were produced from natural SOL through green synthesis. These films with and without PEO polymer were deposited onto glass substrate (at 85 oC for 16 h) via the chemical bath deposition (CBD). The samples were characterized by UV-vis, PL, FESEM, XRD and TEM measurements. The SOL surfactant (as a capping agent)-passivated films without and with PEO polymer displayed improved optoelectronic traits and reduced direct carriers (e^-/h^+) recombination which were ascribed to the strong quantum confinement effect (QCE). FESEM images of the films revealed nanoyarn and nanomat resembling morphologies. XRD patterns of the samples exhibited the existence of ZnO nanocrystallites with (100), (002) and (101) growth planes. The annealed film (at 120 oC) showed an optical band gap of 1.54 eV. TEM micrographs confirmed the nucleation of ZnO NPs size around 4nm and size distribution at 1.2 nm of ZnO QDs. Based on the obtained results it is established that proposed SOL extracts-passivated nanofilm layer may be useful for the photovoltaic applications.

The following comments are as below:

1.      It is strongly recommended that the authors should mention clearly the newly developed and /or found point of in section introduction, compared with papers already reported in this field, adding the references is recommended.

2.      The composition of ZnO nanofilm and ZnO QDs should be checked, e.g., by EPMA or XPS with oxygen vacancy distribution.

3.      The authors reported that a broad emission with peak at 492 nm corresponded to the optical band gap energy of 3.34 eV. The absorption peak was located at 395 nm. In the absence of PEO, the colloidal ZnO NPs revealed a broad emission band at 494 nm with band gap energy of 2.98 eV.

Due to the ZnO is a general material for photocatalysis among the metal oxides, it reacts under UV light due to its large band gap of 3.37 ev, high exciton binding energy (60mev) and optically transparent for visible light. So the absorption spectra of ZnO NPs/SOL and ZnO NPs/SOL-PEO shown in Fig 6 should be checked.

4.      The Emission and absorption spectra of (a) ZnO NPs/SOL-PEO (a') Absorbance (a'') band gap energy, (a''') Emission. (b) ZnO NPs/SOL nanofilms (b') Absorbance, (b'') band gap energy, (b''') Emission shown in Fig 11 should be checked. The typical band gap of ZnO phase is around 3.37 eV.

5.      Shown in Fig 13 should be checked. The typical band gap of ZnO phase is around 3.37 eV. The wrong typing in  (a''), such as Eg= 2.36nm.

6.      Shown in Fig 15 should be checked.

7.      How about the Raman spectra? That could confirm the ZnO phase.

  1. I recommend the authors would show originality in your work compared to other research.
  2. Addition of some discussion compared with above comments is recommended.

Minor editing of English language required.

Author Response

reply to reviewer 5

Round 2

Reviewer 1 Report

The authors have taken into account my comments and have sufficiently improved the reader´s understanding of the paper. However some slight shortages remain:

- Figures 14 and 16 give evidence of large tails in the band structure. Therefore, the derived by the TAUC formula bandgaps are not very reliable. This should be noticed by the authors.

- Formate subscript of TiO2 in Ref.3, 5,12, 16, 35, 37 and of Ti(OH)4 in Ref.5

- Check authors (Udaykumar may be one name) and formate paper title correctly in Ref. 30 and Journal title correctly in Ref. 43,

Now, the paper may be published after minor revision.

Author Response

The answer to reviewer 1 attached 

Reviewer 2 Report

Accept in present form

Author Response

thanks for your comments 

Reviewer 3 Report

The paper can be published in present form.

Author Response

thanks for your efforts 

Reviewer 4 Report

Although I still believe that this paper could benefit from some improvements, the authors have made comprehensive efforts to address all the questions/comments and have made revisions in the resubmitted manuscript. In my opinion, I think the manuscript might be suitable for publication.

Moderate editing of English language required.

Author Response

thanks for your comments 

Reviewer 5 Report

Table (3) represents optical band gap energies of the nano films fabricated. Fig.18 represent Optical band gap energy (Eg) variant with ZnO NPs nanofilm. As depicted in Fig.8, the PL results indicate that SOL phytochemicals have a significant effect on the emission on the suspended ZnO NPs colloidal in the blue-green extended to the yellow orange region, which is due to the defects. The main emission at around 380 nm is also existed, not disappeared. The typical band gap of ZnO phase is between 3.1 and 3.4 eV.

The authors should give a good explanation.

Author Response

Answer to reviewer 5 attached 

Round 3

Reviewer 5 Report

I think that the manuscript can be accepted if other reviewers have no questions.